# Structure and Function of the *ABCD1* Variant Database: 20 Years, 940 Pathogenic Variants, and 3400 Cases of Adrenoleukodystrophy

**DOI:** 10.3390/cells11020283

**Published:** 2022-01-14

**Authors:** Eric J. Mallack, Kerry Gao, Marc Engelen, Stephan Kemp

**Affiliations:** 1Department of Pediatrics, Division of Child Neurology, Weill Cornell Medical College, NewYork-Presbyterian Hospital, New York, NY 10065, USA; ejm9009@med.cornell.edu (E.J.M.); keg4005@med.cornell.edu (K.G.); 2Department of Pediatrics, Memorial Sloan Kettering Cancer Center, New York, NY 10065, USA; 3Department of Pediatric Neurology, Emma Children’s Hospital, Amsterdam University Medical Centers, Amsterdam Neuroscience, University of Amsterdam, 1105 AZ Amsterdam, The Netherlands; m.engelen@amsterdamumc.nl; 4Laboratory Genetic Metabolic Diseases, Department of Clinical Chemistry, Amsterdam University Medical Center, Amsterdam Gastroenterology Endocrinology Metabolism, University of Amsterdam, 1105 AZ Amsterdam, The Netherlands

**Keywords:** adrenoleukodystrophy, peroxisome, ABC transporter, newborn screening, genetics, diagnosis, mutation, variants of uncertain significance, *ABCD1*

## Abstract

The progressive neurometabolic disorder X-linked adrenoleukodystrophy (ALD) is caused by pathogenic variants in the *ABCD1* gene, which encodes the peroxisomal ATP-binding transporter for very-long-chain fatty acids. The clinical spectrum of ALD includes adrenal insufficiency, myelopathy, and/or leukodystrophy. A complicating factor in disease management is the absence of a genotype–phenotype correlation in ALD. Since 1999, most *ABCD1* (likely) pathogenic and benign variants have been reported in the *ABCD1* Variant Database. In 2017, following the expansion of ALD newborn screening, the database was rebuilt. To add an additional level of confidence with respect to pathogenicity, for each variant, it now also reports the number of cases identified and, where available, experimental data supporting the pathogenicity of the variant. The website also provides information on a number of ALD-related topics in several languages. Here, we provide an updated analysis of the known variants in *ABCD1*. The order of pathogenic variant frequency, overall clustering of disease-causing variants in exons 1–2 (transmembrane domain spanning region) and 6–9 (ATP-binding domain), and the most commonly reported pathogenic variant p.Gln472Argfs*83 in exon 5 are consistent with the initial reports of the mutation database. Novel insights include nonrandom clustering of high-density missense variant hotspots within exons 1, 2, 6, 8, and 9. Perhaps more importantly, we illustrate the importance of collaboration and utility of the database as a scientific, clinical, and ALD-community-wide resource.

## 1. Introduction

The *ABCD1* gene is located on Xq28, covers 19.9 kb and contains 10 exons [1,2]. It encodes the ABCD1/ALD protein of 745 amino acids, an ATP-binding cassette transmembrane half-transporter required for the import of coenzyme A-activated very-long-chain fatty acids (VLCFAs; >C22:0) into peroxisomes for degradation [3,4]. Pathogenic variants in the *ABCD1* gene lead to the accumulation of VLCFAs in plasma and tissues, including adrenal glands, testes, the central nervous system, and the subsequent development of X-linked adrenoleukodystrophy (ALD) [5]. ALD has an estimated birth prevalence of 1 in 16,000 births [6,7]. The disease is characterized by a striking clinical spectrum and is unpredictable in disease course and severity, with no established genotype–phenotype relationship [8,9]. About 50% of affected boys will develop primary adrenal insufficiency by the age of 10 [10]. In childhood, 30–35% of boys will develop a fatal cerebral inflammatory disease (cerebral ALD) with a 50–60% total lifetime risk of developing cerebral disease [6,11,12]. In adulthood, virtually all male patients and >80% of affected women develop a chronic progressive myelopathy over their lifetime [10,13,14,15].

Since 1999, most *ABCD1* (likely) pathogenic and benign variants have been reported in the ALD mutation database. The database was initiated as a collaborative effort between the Peroxisomal Diseases Laboratory at the Kennedy Krieger Institute led by Hugo Moser, MD, and the Laboratory Genetic Metabolic Diseases at the Amsterdam UMC in the Netherlands, the initial findings of which were published in 2001 [16]. The aims of the database were to facilitate *ABCD1* variant analysis, collect and catalogue variants in the *ABCD1* gene, improve the analysis of pathogenic variants identified in ALD, facilitate the reporting of novel variants, and make the information overall more accessible by maintaining the database on the internet (original URL: www.x-ald.nl) [16]. The primary aim of this study was to provide an updated analysis of the known variants in *ABCD1* as catalogued over the past 20 years. In addition, we aimed to highlight the database as a resource for physicians, scientists, and patient families; to detail the role of the database in the era of newborn screening for ALD; and, perhaps most importantly, to encourage widespread collaboration, use, and continued data transparency of the database.

## 2. *ABCD1* Variant Database

### 2.1. ABCD1 Variant Analysis, 2001: Initial Results

The first iteration of the ALD Mutation Database [16] reported four polymorphisms and 406 total mutations. Of the 406, 234 (58%) were non-recurrent and 47 were previously unpublished novel pathogenic variants. Missense mutations constituted the most frequent pathogenic variants (55.9%), followed by frame shift (27.1%), nonsense (9.1%), small amino acid insertions/deletions (3.9%), and large deletions (3.9%).

Pathogenic variants in the *ABCD1* gene clustered most often in the transmembrane domain (40%, exons 1 and 2), followed by the ATP-binding domain (30%, exons 6–9), and exon 5 (14%). Non-nested amplification of all exons in the *ABCD1* gene using the Xq28 optimized primer sets developed by Boehm et al. [17] revealed 67% of known mutations clustered into four amplicons covering 40% of the coding region: amplicon 1b in exon 1 (amino acids (aa) 75–188; 20%); amplicon 1c in exon 1 (aa 177–300; 20%); amplicon 8/9 (exons 8 and 9, aa 594–664; 14%); and amplicon 5 (exon 5, aa 465–496; 13%) [17]. A pathogenic variant hotspot was identified in exon 5 accounting for 10.3% of all mutations in the database, c.1415_16delAG (p.Gln472Argfs*83), resulting in a premature stop codon at position 554 [18].

Analysis of missense pathogenic variants revealed a non-random distribution across the gene. Missense mutations were found at higher-than-expected frequency in exon 1 (amino acids 100–300), which code for four segments of the transmembrane domain, and at higher-than-expected frequency in exons 6–9 (amino acids 497–664), which code for the ATP-binding domain.

### 2.2. Evolution of the Database: Experimental Data and Variant Reclassification

The identification of the *ALD* gene in 1993 enabled large-scale genetic testing by diagnostic laboratories and research groups. Between 1993 and 1999, over 40 large-scale pathogenic variant screening studies were published. Unfortunately, two different nucleotide numbering systems were used to annotate variants. The first used the first nucleotide of the cDNA as number +1. The second, and correct system, uses the A of the initiator methionine (ATG) codon as +1. Early versions of the ALD Mutation Database included both numbering systems for each variant. From 2003 onward, all variants have been reported according to the correct numbering system only. Due to this discrepancy, it may be difficult to align newly identified pathogenic variants with cases from those early reports in which the wrong numbering system was used as the nucleotide numbers do not match: they are off by 386. Therefore, reported pathogenic variants from early publications may need to be realigned against the second numbering system. For example, the correct annotation of C696T, R104C [19] is (696 − 386 = 310) c.310C > T, p.Arg104Cys. The *ABCD1* Variant Database reports all (likely) pathogenic and benign variants in conformity with the nomenclature recommended by the Human Genome Variation Society (HGVS; https://varnomen.hgvs.org/) [20]. All variants, including those already published, are annotated using the Alamut Visual software package with transcript NM_000033.3 on GRCh37 (hg19) as the reference sequence.

In 2017, following the expansion of ALD newborn screening in the U.S., the *ABCD1* Variant Database became part of Adrenoleukodystrophy.info. This name better reflects the broader focus on ALD. The website also moved to a new domain name: https://adrenoleukodystrophy.info/.

The *ABCD1* Variant Database was also rebuilt. Instead of merely reporting whether a variant was already reported in literature or shared by diagnostic laboratories, the new version contains additional information: it lists the number of cases reported for each pathogenic variant. An ALD case is defined as an individual with clinical signs and symptoms related to ALD and a biochemical or genetic confirmation. Where available in the literature, experimental data were extracted supporting the pathogenicity of a particular variant (a Western blot, peroxisomal beta-oxidation, etc.). For example, the pathogenic variant p.Ser606Leu has been reported in 37 ALD cases. Other ALD-related research studies using patient cell lines revealed that this pathogenic variant reduced the amount of detectable ALD protein to 25% that of control cells (Database Ref 97) [21]. Studies using experimental cell lines showed that the pathogenic variant affects ATP-binding capacity (Database Refs 194 and 244) [22,23], resulting in deficient VLCFA beta-oxidation (Database Ref 261) [24].

Information from publicly available databases, such as the genome Aggregation Database (gnomAD, https://gnomad.broadinstitute.org/), was also included [25]. In some instances, a variant that was reported to be pathogenic could be correctly reclassified as benign. For example, p.Gly608Asp was reported as the pathogenic variant in an ALD male patient (Database Ref 48) [26], but data from gnomAD show its frequency to be 124/162246 alleles (X:153008483 G/A), confirming that this is not a pathogenic variant.

In addition to the *ABCD1* Variant Database, the website provides information on a number of ALD-related topics. The website pages are written by ALD researchers and physicians with expertise on that topic. A number of these pages are available in Spanish, French, German, and Dutch. Interestingly, some of the most visited pages are in Spanish, which indicates that there is an unmet need for information written by experts in other languages. This motivates us to continue adding translations; however, this is a relatively slow process as the database is maintained without funding. Hence, we are fully dependent on collaborators with a biology background/medical training willing to translate content. Parallel to the expansion of ALD newborn screening, there has been a steady increase in visitors with an annual average of 30,000 visits before 2014 to over 150,000 since 2019. There also has been a shift in the devices that visitors use to visit the website. As of 2020, the majority of visitors have used a mobile phone to visit the website.

### 2.3. ABCD1 Variant Analysis, 2021: Materials and Methods

*ABCD1* variant data were analyzed and annotated in RStudio (v.1.2.5033) using the Bioconductor package trackViewer [27,28]. Dandelion plots were generated to depict the distribution and density of variants by chromosome position across the *ABCD1* gene according to assembly GRCh37 (hg19) (Figure 1). The height of each dandelion reflects the density of variants at that position. The size of the chromosome visualized per plot was controlled by coding the ratio of the chromosome segment size to the maxgaps variable: a ratio of 8 for visualization of the whole gene (19,893 bp), 5 for visualization of the open reading frame of exon 1 in its entirety (900 bp, Figure 2b), and 2 for comparison across comparably sized segments of the gene (100–300 bp each): exons 2, 6, 8, and 9. A multiple sequence alignment of ABCD1 proteins from 54 different species (from *Danio rerio* to human) was created using ClustalW [29,30] (see Appendix A for the multiple sequence alignment). The WebLogo program (http://weblogo.threeplusone.com/) was used to generate a graphic representation of the evolutionary conservation of each amino acid within the ABCD1 protein [31].

### 2.4. ABCD1 Variant Analysis, 2021: Results

As of December 2021, the *ABCD1* Variant Database had collected 3401 cases representing 948 non-recurrent, unique, (likely) pathogenic variants as well as 40 benign variants and 249 variants of uncertain significance (VUS). Missense pathogenic variants continue to account for the most frequent variants in the entire dataset (61.4%), followed by frame shift (17.2%), nonsense (9.9%), splice site (4.3%), small amino acid insertions/deletions (3.5%), large deletions (2.6%), and benign variants (1.2%). The distribution is generally consistent across the 948 non-recurrent variants (Table 1).

Pathogenic variants in *ABCD1* cluster most often in the transmembrane domain (46%, exons 1–2), followed by the ATP-binding domain (35%, exons 6–9), followed by exons 3 and 4 (9%) and exon 5 (7%). The most frequent pathogenic variant continues to be the 2 bp deletion in exon 5 p.Gln472Argfs*83 (*n* = 170, 5.0% of all pathogenic variants), followed by missense pathogenic variants p.Arg554His (*n* = 70, 2.1%), p.Arg660Trp (*n* = 60, 1.8%), p.Arg617His (*n* = 57, 1.7%), and p.Arg518Gln (*n* = 54, 1.6%). All pathogenic variant types are homogenously distributed spatially across *ABCD1*. Eighty-nine large exon deletions are reported, also distributed across the entire gene, the most frequent of which are exon 8–10 del (*n* = 21), exon 7–10 del (*n* = 16), and exon 3–10 del (*n* = 12). Entire *ABCD1* gene deletions have also been reported. However, complete deletion of *ABCD1* extends into the promoter region of the upstream neighboring gene (*BCAP31*). The combined deletion or dysfunction of both *ABCD1* and *BCAP31* (initial name *DXS1375E* [32]) causes contiguous *ABCD1 DXS1357E* deletion syndrome (CADDS). CADDS is characterized by severe intrauterine growth retardation, neonatal hypotonia, failure to thrive, severe global developmental delay, and liver dysfunction, leading to early death [33].

The density of all variants (pathogenic, benign, synonymous, and VUS combined) in the database are visualized in Figure 1. Overall, variant density is highest in exon 6, followed by exon 1 (Figure 1a). The highest density of pathogenic variants occurs in exon 1 followed by exon 6, with comparable densities in exons 2, 8, and 9 (Figure 1b). Exon 1 contains the overall highest pathogenic variant burden distributed in the latter 2/3 of the exon (amino acids 100–300), studded with multiple high-density hotspots (Figure 1c). Benign variants and variants of uncertain significance (VUS) are homogenously distributed across *ABCD1* with no variation in density (data not shown).

The density of missense variants across the whole gene and across multiple exons is plotted in Figure 2. Missense variants cluster across chromosome segment 152991000–152991621 (exon 1, amino acids 100–300, Figure 2b), and in the chromosome segment 152994751–152994822 (exon 2, amino acids 315–346, Figure 2c), which encodes the transmembrane domain. Notable high-density missense hotspots are located in the ATP-binding domain encoded by segments in exon 6 (chromosome position 153005575–153005610, amino acids 497–545, Figure 2d), exon 8 (chromosome position 153008470–153008520, amino acids 604–620, Figure 2e), and in exon 9 (chromosome position 153008685–153008710, amino acids 623–664, Figure 2f).

## 3. *ABCD1* Variant Interpretation: 2001 vs. 2021

The order of (likely) pathogenic variant frequency, overall nonrandom clustering of disease-causing variants in exons 1–2 (transmembrane domain) and exons 6–9 (ATP-binding domain), and the most commonly reported pathogenic variant p.Gln472Argfs*83 on exon 5 are consistent with the initial reports of the mutation database [16]. A few novel insights are afforded by the updated analysis and are secondary to the ability to map variants onto the gene with higher resolution. The clusters of missense variants in exons 6 and 9 are not equally distributed; rather, they form high-density hotspots in both exons. Given that VUS, synonymous, and benign variants appear to be equally distributed across all exons (Figure 1a), these observations imply that: (1) these segments of the protein are intolerant to variation, and (2) a VUS that occurs in either of the hotspots on exon 6 or 9 may have a higher likelihood of being pathogenic. The same logic applies to VUSs that appear in exon-1-segment amino acids 100–300, chromosome interval 152994751–152994822 (amino acids 315–346) on exon 2, and exon 8 (amino acids 594–622). At the functional level, the high-density hotspots on exons 6 and 9 identified independently via the bioinformatics analysis of missense variants map to two specific motifs within the ATP-binding domain: Walker A and B and the ABC signature (Figure 2d–f) [4]. Early experimental data revealed that the helical domain between the Walker A and B motifs undergoes conformational change when bound to ADP, indicating their role in the transduction of free energy during ATP-hydrolysis in the ALD protein [34,35]. Further work demonstrated the role of the Walker A and B motifs in forming the ATP bindings site sandwich required for homodimerization of the ALD protein, and the subsequent power stroke of ATP-dependent substrate transport [36,37]. This highlights the potential for accurate variant curation and analysis of the database to identify functionally important segments of the gene (Figure 3). In the case of exons 6, 8, and 9, the functional data provide further validation that novel variants in these high-density clusters have a higher likelihood of being pathogenic.

## 4. *ABCD1* Variant Interpretation in the Era of ALD Newborn Screening

### 4.1. Newborn Screening for ALD

Adrenal crises and the onset of cerebral ALD confer significant morbidity and mortality in childhood without proper disease monitoring and treatment [7,38]. Before the age of 10 years, one out of two boys diagnosed with ALD will develop adrenal insufficiency and one out of three boys will develop cerebral ALD [8,10]. Historically, boys who presented with neurologically symptomatic cerebral ALD translated to extensive disease on MRI and poor hematopoietic stem cell transplant outcomes [11,39,40,41,42]. Early-stage treatment of childhood cerebral ALD with hematopoietic stem cell transplantation and, more recently, gene therapy can arrest disease progression if performed in the window of opportunity when the MRI is abnormal but neurological symptoms are not yet apparent [7,38,43,44,45,46,47,48,49,50]. The implementation of newborn screening for ALD in the United States and the Netherlands has provided the opportunity to identify patients at birth [7,51,52,53,54,55,56]. Asymptomatic boys are monitored for adrenal insufficiency [57] and by MRI [58] to detect early cerebral lesions, with the aim of intervention in the narrow window prior to symptom onset [41,57,58].

### 4.2. Asymptomatic Diagnosis Requires a Platform to Resolve Variants of Uncertain Significance

Newborn screening gives rise to the clinical situation where patients, without a clear family history of disease, are identified asymptomatically at birth with a novel variant or VUS. This is clearly different from the more classical situation prior to newborn screening where index patients presented with signs and symptoms of disease, and a diagnosis was made based on confirmatory biochemistry and genetics. A family history would then be taken to identify other family members at risk of ALD, or to diagnose other symptomatic family members. In newborn screening, or population screening, patients are also identified that have C26:0-lysoPC levels above the upper level of the reference range, but below the lower level of the disease range. The identification of a novel variant or VUS creates a dilemma: is the VUS a benign change or a pathogenic variant? The diagnosis of index cases identified by newborn screening requires the demonstration of elevated VLCFAs and an accurate interpretation of the genetic variant in *ABCD1*. In the case of a VUS, biochemical and functional analyses of patient fibroblasts can aid the interpretation of the variant as either a (likely) benign or pathogenic change [59].

Taken together, the *ABCD1* Variant Database, open collaboration, and the results of this study serve as a platform to aid in the resolution of VUSs identified during newborn screening. This is in addition to other publicly available variant archives (e.g., ClinVar: https://www.ncbi.nlm.nih.gov/clinvar/; Human Genome Variation Society: http://www.hgvs.org/locus-specific-mutation-databases). Medical professionals can report the biochemical evidence or functional studies supportive of a VUS’s pathogenicity, map the variant onto the gene to visualize whether it exists within a pathogenic variant cluster or hotspot, and assess the conservation of the potentially affected amino acid (Figure 3). For example, the VUS c.970C > T (p.Arg324Cys) within the exon 2 missense cluster was resolved in this way. The patient developed elevated VLCFAs and an elevated ACTH, was diagnosed with partial adrenal insufficiency before 2 years old, and is now proactively treated with stress dose steroids. Moving forward, the database provides a wealth of input to understand the effect of missense variants on protein structure and folding, as emerging presently with neural-network-based protein modeling such as AlphaFold [60].

### 4.3. The Clinical and Collaborative Importance of the ABCD1 Variant Database

The facilitation of a correct interpretation of a variant’s pathogenicity in *ABCD1* is highly important. The accurate genetic diagnosis of ALD obligates the patient to rigorous monitoring protocols for adrenal insufficiency [57] and cerebral ALD by MRI [58], requiring multiple medical subspecialty evaluations early in life. If an early-stage cerebral lesion is identified in a boy with neurologically asymptomatic ALD [61], he will be referred for hematopoietic stem cell transplantation or gene therapy, if applicable. The former is associated with its own degree of morbidity and mortality [62], and both require the patient undergo chemotherapy [48]. Lastly, the identification of a familial gene mutation may lead to a life-altering diagnosis of ALD in extended family members.

Despite ALD being an X-linked disorder, a genetic diagnosis is especially important for women. It is becoming increasingly recognized that the majority of women are not merely carriers, but may develop some degree of myeloneuropathy leading to gait, bowel, and bladder dysfunction [14,15,63]. In terms of family planning, identification of index boys in new families with newborn screening has uncovered the expected maternal inheritance pattern of the pathogenic variant. This has caused young families to pursue prenatal genetic diagnosis and in vitro fertilization of unaffected embryos. This process, while necessary to prevent further propagation of the disease to future generations, imposes significant medical, emotional, and financial distress on new families trying to have more children [64].

From a collaborative standpoint, the downstream effects have been multiple. Updating relevant clinical data to novel VUSs and pre-existing reported missense pathogenic variants has allowed physicians across the world to network quickly. This was the case for two newborn screening patients born in two different states who were identified with the same reported VUS c.970C > T (p.Arg324Cys) above. Rapid reporting of pathogenicity and sharing of information via the ABCD1 variant database have enhanced the care for the patients, their families, and their treating physicians. Lastly, the database has allowed families to take an active part in their child’s care by being a resource for researching their child’s variant and learning about ALD through well-curated information.

## 5. Conclusions

A unique resource such as the *ABCD1* Variant Database can only be maintained through open collaboration and the sharing of (de-identified) information. If well-maintained, it helps the ALD community at large. From a scientific perspective, the database provides a backbone for realignment of older reported variants, a resource for the integration of experimental data and large datasets, a platform for variant classification, and a way to shed light on functionally important segments of the protein. Clinically, it helps to address challenges posed by novel variants in the setting of newborn screening. For the ALD community, the database provides an opportunity to collaborate, translate, and transmit well-curated disease information to both non-expert physicians and families affected by ALD.

## Figures and Tables

**Figure 1 cells-11-00283-f001:**
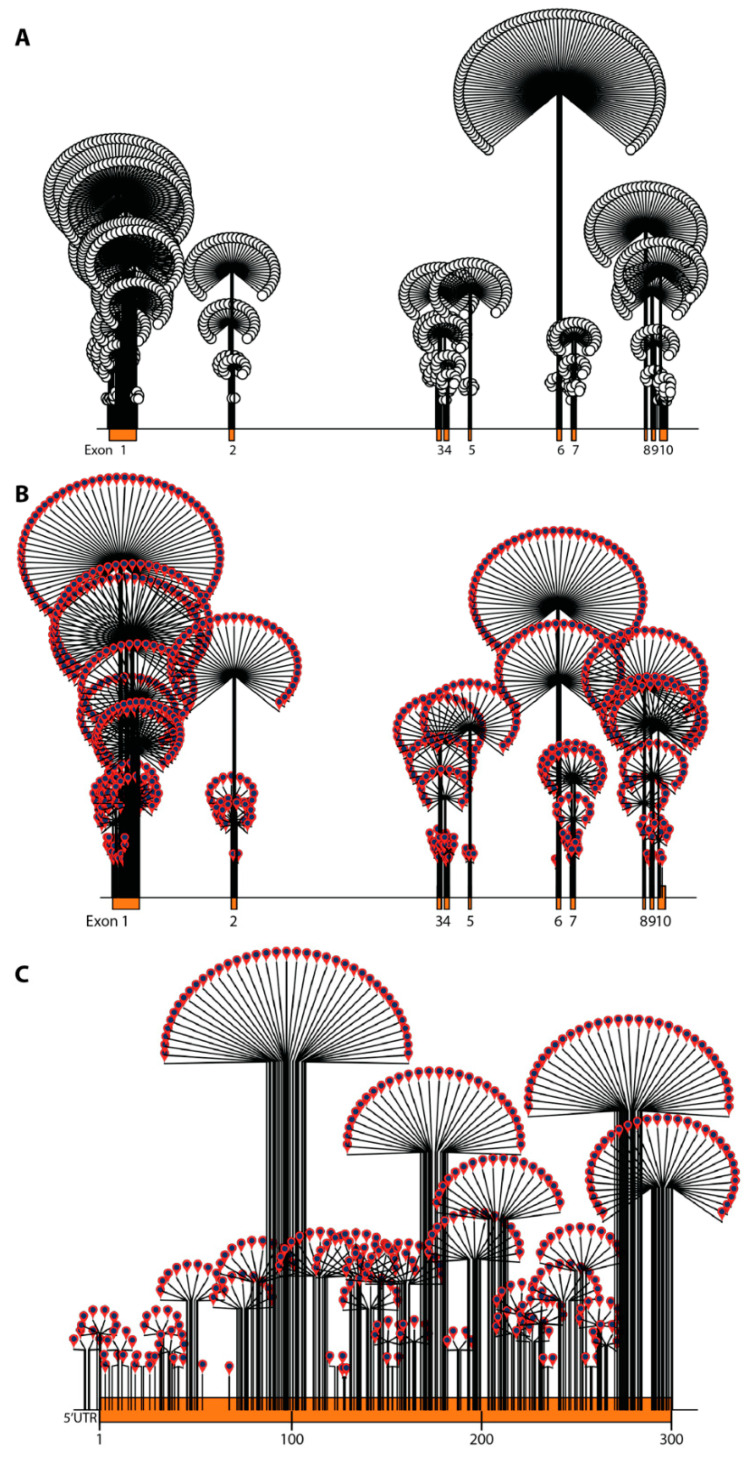
Dandelion plot illustrating variant density in *ABCD1* (density is indicated by height of dandelion). (**A**) All variants (pathogenic, benign, VUS, and synonymous) in the *ABCD1* gene (open circles). The highest variant density is in exon 6 followed by exon 1. (**B**) Only pathogenic variants in the *ABCD1* gene (red pins) are shown. The highest event burden and pathogenic variant density are in exon 1 followed by exon 6. (**C**) All pathogenic variants in exon 1.

**Figure 2 cells-11-00283-f002:**
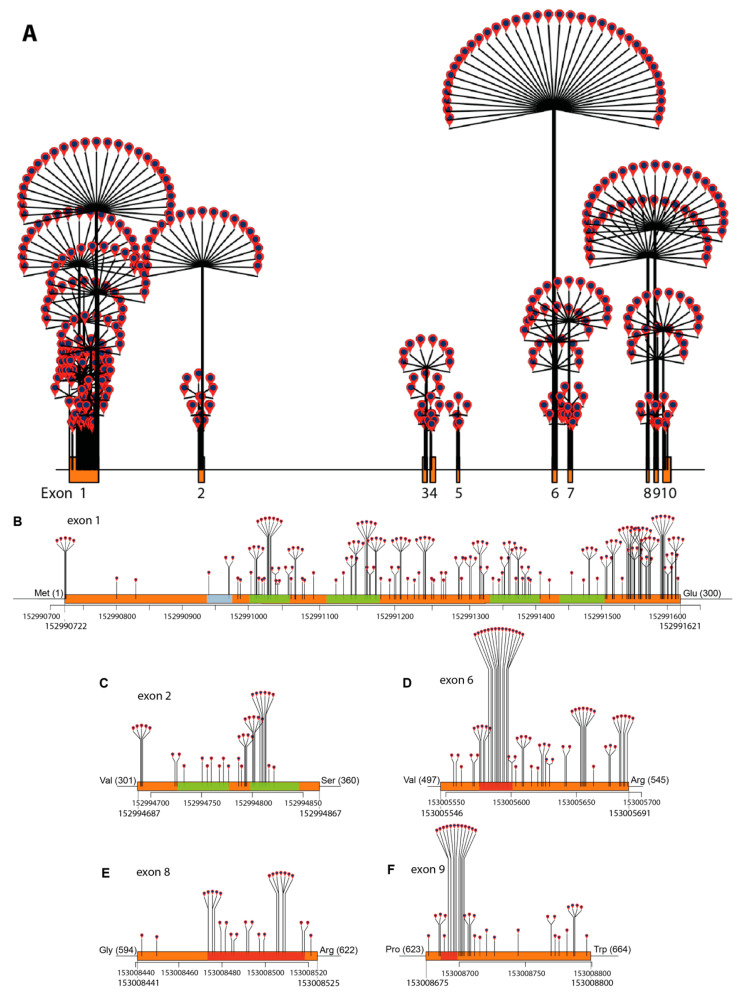
Missense variant density by chromosome position: (**A**) across the *ABCD1* gene, (**B**) exon 1, (**C**) exon 2, (**D**) exon 6, (**E**) exon 8, and (**F**) exon 9. Exons are indicated in orange, the PEX19-binding site is indicated in blue, the transmembrane segments are indicated in green and Walker A and B and the ABC signature are indicated in red.

**Figure 3 cells-11-00283-f003:**
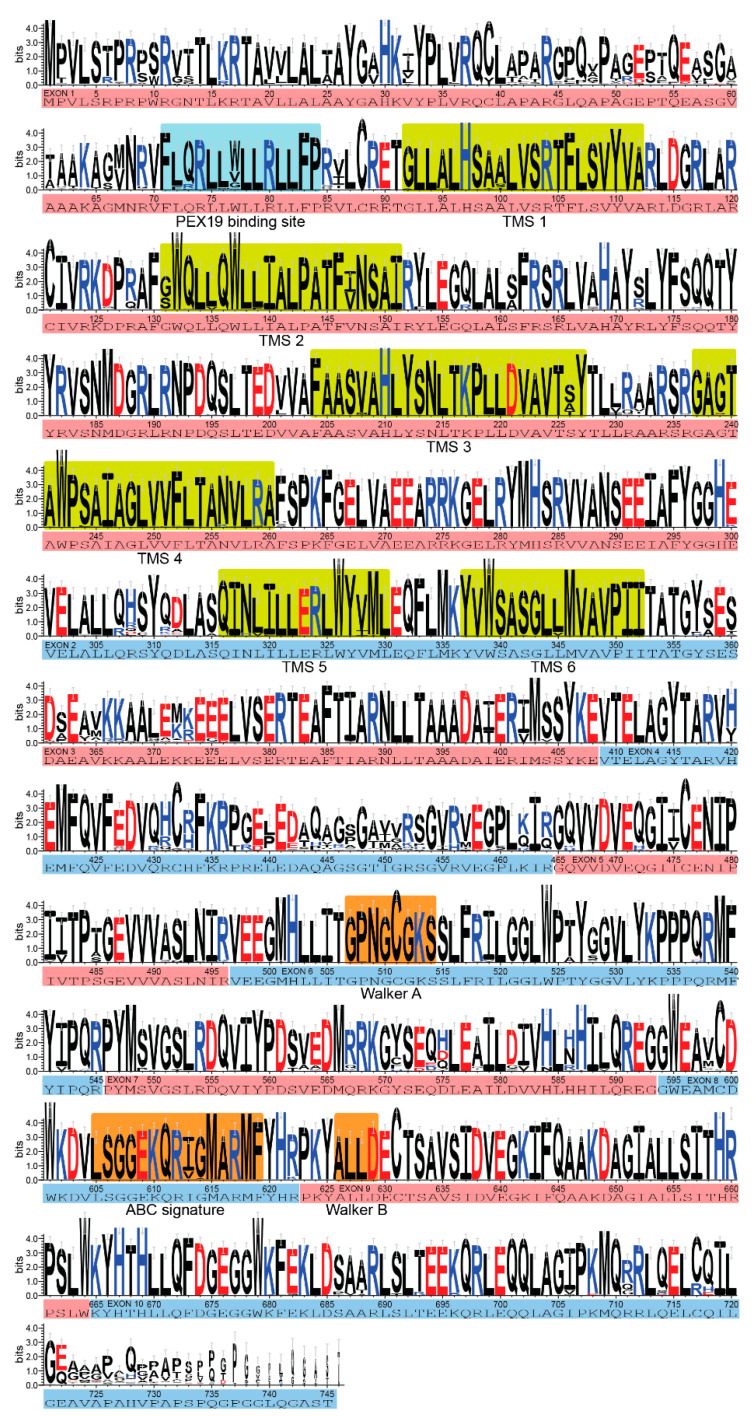
The evolutionary conservation of each amino acid within the ABCD1 protein based multiple sequence alignment among 54 different species. The overall height of each symbol represents the relative frequency of each amino acid at that position. Positively charged amino acids (KRH) are in blue; negatively charged amino acids (DE) are in red. Exons are indicated in alternating blue–pink–blue, etc.; the PEX19-binding site is indicated in blue; the 6 transmembrane segments are indicated in green; Walker A and B and the ABC signature are indicated in orange. Error bars indicating an approximate Bayesian 95% confidence interval are indicated in grey.

**Table 1 cells-11-00283-t001:** Variant Counts and Frequencies in *ABCD1* variants.

	Total		Unique	
	*n*	%	*n*	%
All *ABCD1* variants in the database	3401		948	28%
Missense pathogenic variants	2087	61.4%	411	43.4%
Nonsense pathogenic variants	336	9.9%	116	12.3%
Frame shift pathogenic variants	585	17.2%	262	27.6%
Amino acid insertions/deletions	119	3.5%	52	5.5%
Splice site pathogenic variants	145	4.3%	43	4.5%
One or more exons deleted	89	2.6%	24	2.5%
Benign variants	40	1.2%	40	4.2%

## Data Availability

Following publication, any data not published within this article will be shared on request from any qualified investigator.

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
