# Peer review of "Structure and Function of the ABCD1 Variant Database: 20 Years, 940 Pathogenic Variants, and 3400 Cases of Adrenoleukodystrophy"

_cells, 2022, doi:10.3390/cells11020283_

Round 1

Reviewer 1 Report

Mallack and colleagues report on a  large database of  adrenoleukodystrophy mutation 2 mutations, an important subject in its field.

The manuscript is well written and timely.

Minor comments include:

  • Perhaps the title should include  reference to the relevant gene: 

    the ABCD1 gene.

  • The motion of other languages is intriguing: what do the authors propose to do about the matter they raised (p.3)?
  • The mention of VUS is important and deserves further discussion (p. 9).
  • It might be worthwhile mentioning the paper of Higgins et al. (2021) on DNA nomenclature in the Discussion. This paper addresses a thorny issue plaguing much of human genetics and genomics which databases should address.

Author Response

Comment 1: Perhaps the title should include reference to the relevant gene: the ABCD1 gene.

Response: We thank the reviewer for this suggestion. Also, in response to the comment made by reviewer 2 about the nomenclature we have changed the title to: “Structure and function of the ABCD1 variant database: 20 years, 940 pathogenic variants and 3400 cases of adrenoleukodystrophy”

Comment 2: The motion of other languages is intriguing: what do the authors propose to do about the matter they raised (p.3)?

Response: We believe the matter referred to in this reviewer’s comment is “Interestingly, some of the most visited pages are in Spanish, which indicates that there is an unmet need for information written by experts in other languages.” We added the following sentence (lines 133-136) “This motivates us to continue adding translations. However, this is a relatively slow process as the database is maintained without funding. Hence, we are fully dependent on collaborators with a biology/medical training willing to translate content.”

Comment 3: The mention of VUS is important and deserves further discussion (p. 9).

Response: We have added the statement and reference of the recent work published by van de Stadt and colleagues on functional experiments aimed at resolving VUSs in ABCD1 (lines 269-271); “In the case of a VUS, biochemical and functional analyses of patient fibroblasts can aid the interpretation of the variant as either a (likely) benign or pathogenic change [59].”

Comment 4: It might be worthwhile mentioning the paper of Higgins et al. (2021) on DNA nomenclature in the Discussion. This paper addresses a thorny issue plaguing much of human genetics and genomics which databases should address.

Response: Thank you for ensuring that we follow the agreed upon HGVS guidelines.  Variant formatting was indeed updated. In line with the statement provided on the database (online, https://adrenoleukodystrophy.info/mutations-and-variants-in-abcd1) we added the phrase (with the appropriate citation) to page 3 under section 2.2 (lines 101-106): “The “ALD Variant Database” reports all (likely) pathogenic and benign variants conform to the nomenclature recommended by the Human Genome Variation Society (HGVS; https://varnomen.hgvs.org/)[20]. All variants, including those already published, are annotated using the “Alamut Visual” software package using transcript NM_000033.3 on GRCh37 (hg19) as reference sequence.”.

Reviewer 2 Report

Kemp and colleagues report the status of their ALD variant database, which has been available online since 1999. Their database is used by the X-ALD community and is very useful and a great benefit to the community. I only have a few suggestions, primarily on using accepted genetics terminology and providing definitions.

Comments:

  1. Throughout: Per the ACMG and the genetics community, the terms ‘mutation’ and ‘polymorphism’ are discouraged due to having multiple conflicting meanings that can be subjective. Throughout the manuscript, when describing variants, please use recommended terminology throughout the manuscript: ‘variant’, and if applicable or necessary, add one of the 5 qualifiers: ‘pathogenic’, ‘likely pathogenic’, ‘VUS’, ‘likely benign’, ‘benign’.
  2. Throughout: VUS = variant of uncertain significance, not variant of unknown significance.
  3. Please provide the definition of how a case is defined in the database. Are all ALD phenotypes considered cases? When a lab submits a variant detected in a case, are there requirements or guidelines for case phenotype, biochemical testing or genetic analysis? If none, this should be stated. 
  4. As the authors highlight on page 9, line 275, ‘…correct interpretation of a variant’s pathogenicity in ABCD1 is highly important.’ Please provide a description of who classifies and assigns pathogenicity for variants in the database – is it the submitter, or the database team, or both? Are there requirements on how variants should be classified at submission?
  5. Variants are stratified into one of 4 groups using a mixture of classification (pathogenic, VUS, benign) and type (synonymous). Is this because synonymous variants are not classified due to difficulty in interpretation? This makes sense, please just state the rationale for separating them.
  6. Please justify why ‘likely pathogenic’ and ‘likely benign’ aren’t used in the manuscript. Are they not used in the database?  In my experience, there are just as many likely benign as benign variants, for example.
  7. Table 1: The headings could be more informative. Based on the title of the manuscript, does data in the first column marked ‘recurrent N’ count the total number of unique variants detected among cases (?), and the fourth column marked ‘Non-recurrent N’ count the number of unique cases(?).
  8. Figure 3: ABCD1 sequence logo. Please list the reference that the relative frequency was compared against. Presumably, the references are ABCD1 sequences from some number of organisms (please list)?
  9. Do you submit variants to ClinVar, or do submitting labs typically do this?
  10. Suggestion, not required: It would be interesting to note how often the classification provided by your database matches the classification in ClinVar.

Minor/Typos:

  1. Page 7, line 13: There is a missing qualifier. ‘Before the age of 10 years, 1 out of 2 boys …’ Perhaps ‘…with biochemically confirmed ALD…’  or ‘…with a pathogenic ABCD1 variant …’
  2. Throughout: Sometimes (not always), only partial genomic coordinates are provided (see page 6, lines 183-189 as an example). Please always list the entire chromosomal position to avoid confusion.
  3. Please list the reference genome used for coordinates (e.g. hg18, hg19).
  4. Page 9, line 282: ‘…undergo a chemotherapy…’ or ‘…undergo chemotherapy…’ ?
  5. Page 9, line 293: ‘…generations, is imposes…’ to ‘…generations imposes…’?
  6. Page 10, line 295: ‘affects’ should be ‘effects’.

Author Response

Comment 1: Throughout: Per the ACMG and the genetics community, the terms ‘mutation’ and ‘polymorphism’ are discouraged due to having multiple conflicting meanings that can be subjective. Throughout the manuscript, when describing variants, please use recommended terminology throughout the manuscript: ‘variant’, and if applicable or necessary, add one of the 5 qualifiers: ‘pathogenic’, ‘likely pathogenic’, ‘VUS’, ‘likely benign’, ‘benign’.

Response: Thank you for ensuring we are using the correct, and most accurate nomenclature and qualifiers. We have updated this throughout the paper. In the earlier parts of the manuscript, the term “mutation” was left given it was the historical name of the database and the common term used 20 years ago. The reviewers will note that any mention of “mutation” has been updated when referring to the present form of the database or present analyses (overall >80% of the term “mutation” has been replaced in the text to “pathogenic variant” or “variant”). In line with the statement provided on the database (online, https://adrenoleukodystrophy.info/mutations-and-variants-in-abcd1) we added the phrase (with the appropriate citation) to page 3 under section 2.2 (lines 101-106): “The “ALD Variant Database” reports all (likely) pathogenic and benign variants conform to the nomenclature recommended by the Human Genome Variation Society (HGVS; https://varnomen.hgvs.org/)[20]. All variants, including those already published, are annotated using the “Alamut Visual” software package using transcript NM_000033.3 on GRCh37 (hg19) as reference sequence.”.

In addition, where possible we also changed “mutation” to “variant” in the online database (https://adrenoleukodystrophy.info/).

Comment 2: Throughout: VUS = variant of uncertain significance, not variant of unknown significance.

Response: In literature, both “unknown” and “uncertain” are used. We do agree that “uncertain” better captures the underlying problem and have changed this throughout the paper.

Comment 3: Please provide the definition of how a case is defined in the database. Are all ALD phenotypes considered cases? When a lab submits a variant detected in a case, are there requirements or guidelines for case phenotype, biochemical testing or genetic analysis? If none, this should be stated.

Response: It is not really clear what the reviewer means with “Are all ALD phenotypes considered cases?”. We have added a case definition to the section that focusses on rebuilding the database in 2017 (lines 114-115): “An ALD case is defined as an individual with clinical signs and symptoms related to ALD and a biochemical or genetic confirmation”.

The cases that have been submitted by diagnostic labs all came with a clinical description, biochemical and genetic confirmation. Almost all these entries are from >10 years ago. At present, the majority of genetic screening is outsourced to commercial parties (at least in the US) and these companies do not share their information with the database. 

Comment 4: As the authors highlight on page 9, line 275, ‘…correct interpretation of a variant’s pathogenicity in ABCD1 is highly important.’ Please provide a description of who classifies and assigns pathogenicity for variants in the database – is it the submitter, or the database team, or both? Are there requirements on how variants should be classified at submission?

Response: The vast majority of variants are derived from publications. The primary interpretation/classification is made by the researchers/physicians that publish the work. In case conflicting data appears in literature (a second publication with other findings), the database team will include this in the remarks. An example is given in the manuscript (lines 124-128): “In some instances, a variant that was reported to be pathogenic could be correctly reclassified to benign. For example, p.Gly608Asp was reported as the pathogenic variant in an ALD male patient (Database Ref 48) [26], but data from gnomAD shows its frequency to be 124/162246 alleles (X:153008483 G/A) confirming that this is not a pathogenic variant.”. Another example for p.Arg660Gln is available in the database online: “Conflicting results. Reported as pathogenic (3 ALD cases) (71, 72) with no detectable ALDP in patient cells (72). But, an independent functional study in fibroblasts showed normal ALDP and normal biochemistry (290) and normal plasma VLCFA (290).”

Comment 5: Variants are stratified into one of 4 groups using a mixture of classification (pathogenic, VUS, benign) and type (synonymous). Is this because synonymous variants are not classified due to difficulty in interpretation? This makes sense, please just state the rationale for separating them.

Response: It is not really clear where in the manuscript the reviewer likes us to address this comment. We always thought synonymous variants are meaningless. But, is it ever in biology - like the “junk DNA” in the past? It is true that in general we think that a synonymous variant (no amino acid change) is harmless. In a recent paper (van de Stadt et al 2021) we addressed this in relation to a VUS and conflicting results (in fact this relates to the p.Arg660Gln example given in response to comment 4): “Although the effect of synonymous variants in ABCD1 has not been investigated, we cannot rule out that they may affect the mRNA stability and ALDP expression, as has been described in the cases of other proteins [30]. It cannot be excluded that c.1899C>T (p.Ser633Ser) and/or c.1950G>A (p.Ala650Ala) aggravate the deleterious effect of p.Arg660Gln on ALDP expression or stability. Therefore, the presence or absence of such a functionally synonymous variant may determine the pathogenic outcome of a missense variant.”

Comment 6: Please justify why ‘likely pathogenic’ and ‘likely benign’ aren’t used in the manuscript. Are they not used in the database?  In my experience, there are just as many likely benign as benign variants, for example.

Response: Please see our response to comment 1. We have added “likely” throughout the manuscript where it is applicable.

Comment 7: Table 1: The headings could be more informative. Based on the title of the manuscript, does data in the first column marked ‘recurrent N’ count the total number of unique variants detected among cases (?), and the fourth column marked ‘Non-recurrent N’ count the number of unique cases(?).

Response: Thank you for pointing this out.  We updated the headers to be more specific, as this reviewer correctly pointed out.  Recurrent = total.  “Total” has replaced “Recurrent”. Non-recurrent = Unique, and has also been replaced as such in Table 1.

Comment 8: Figure 3: ABCD1 sequence logo. Please list the reference that the relative frequency was compared against. Presumably, the references are ABCD1 sequences from some number of organisms (please list)?

Response: we agree that just mentioning “A multiple sequence alignment of ABCD1 proteins from 54 different species (from Danio rerio to human) was not very informative. We have added the multiple sequence alignment as supplemental data. Lines 150-153: “A multiple sequence alignment of ABCD1 proteins from 54 different species (from Danio rerio to human) was created using ClustalW [29,30] (See supplemental Figure 1 for the multiple sequence alignment).”

Comment 9: Do you submit variants to ClinVar, or do submitting labs typically do this?

Response: We do not submit variants to ClinVar. Many variants in ClinVar come from commercial companies like Invitae, without any supporting information. At present, the majority of genetic screening is outsourced to commercial parties (at least in the US) and these companies do not share their information with the database (even though we know they use the database). We do not know if they all share their findings with ClinVar.

Comment 10: Suggestion, not required: It would be interesting to note how often the classification provided by your database matches the classification in ClinVar.

Response: we haven’t done this systematically. But often ClinVar reports only a single case for a variant. For example, p.Met1Val is listed to be pathogenic in ClinVar (with a single entry), but without any supporting data. The ALD variant database lists 23 ALD cases for this variant with supporting data and references: “Pathogenic, identified in 23 ALD cases (32, 102, 139, 141, 218, 266). No detectable ALDP in patient cells (102, 139, 141).”

Comment 11: Page 7, line 13: There is a missing qualifier. ‘Before the age of 10 years, 1 out of 2 boys …’ Perhaps ‘…with biochemically confirmed ALD…’  or ‘…with a pathogenic ABCD1 variant …’

Response: Thank you! We added the qualifier “diagnosed with ALD” (line 239).

Comment 12: Throughout: Sometimes (not always), only partial genomic coordinates are provided (see page 6, lines 183-189 as an example). Please always list the entire chromosomal position to avoid confusion.

Response: We corrected this in the appropriate occasions within the paper on page 8.

Please list the reference genome used for coordinates (e.g. hg18, hg19).

Response: Thank you for pointing this out. On page 3, section 2.3, line 145 we added “… according to assembly GRCh37 (hg19)”

Page 9, line 282: ‘…undergo a chemotherapy…’ or ‘…undergo chemotherapy…’ ?

Response: We removed “a” from the phrase, so it now reads “…undergo chemotherapy” (line 295)

Page 9, line 293: ‘…generations, is imposes…’ to ‘…generations imposes…’?

Response: We removed “is”, so it now reads “This process, while necessary to prevent further propagation of the disease to future generations, imposes significant medical,…..” (line 304)

Page 10, line 295: ‘affects’ should be ‘effects’ .

Response: Thank you, affects has been replaced with “effects” (line 307).

We hope that we have answered the reviewer comments satisfactorily. We hope that our paper is now acceptable for publication, but stand ready to address further concerns if necessary.